# Usefulness of Routine Laboratory Tests for Follow up of Patients Receiving Outpatient Parenteral Antimicrobial Therapy Run by Infectious Diseases Fellows

**DOI:** 10.3390/antibiotics12020330

**Published:** 2023-02-04

**Authors:** Jared Frisby, Naureen Ali, Samson Niemotka, Getahun Abate

**Affiliations:** Division of Infectious Diseases, Doisy Research Center, Saint Louis University, 1100 S. Grand Blvd., Saint Louis, MO 63104, USA

**Keywords:** OPAT, antibiotic, long term, complication

## Abstract

Background: The infectious disease society of America (IDSA) recommends routine laboratory tests for all patients receiving outpatient parenteral antimicrobial therapy (OPAT) to monitor for adverse events. There are no data to support how often patients should take monitoring laboratory tests. In addition, the relevance of different laboratory tests commonly used for OPAT follow up is not clearly known. Methods: We conducted a retrospective observational cohort study over a 7-year study interval (1 January 2014 to 31 December 2021). Clinical data were obtained to identify the risk factors associated with abnormal laboratory tests and determine if abnormal laboratory tests led to antibiotic change or hospital readmission. Results: Two hundred and forty-six patients met the inclusion criteria for this study. In our multivariate analysis, the Charlson comorbidity index (CCI) of 0–4 (aOR 0.39, 95%Cl 0.18–0.86), the use of ceftriaxone without vancomycin (aOR 0.47, 95%Cl 0.24–0.91) and an OPAT duration of 2–4 weeks (aOR 0.47, 95%Cl 0.24–0.91) were associated with a lower risk of OPAT complications. A CCI of 5 or more (aOR 2.5, 95%Cl (1.1–5.7)) and an OPAT duration of 5 or more weeks (aOR 2.7, 95% Cl 1.3–5.6) were associated with a higher risk of OPAT complications. An abnormal complete metabolic panel or vancomycin levels, but not an abnormal complete blood count, were associated with antibiotic change or readmission. Conclusion: Patients with fewer comorbidities, ceftriaxone and short OPAT durations are at lower risk for OPAT complications. These patients could be followed with less frequent laboratory monitoring.

## 1. Introduction

Outpatient parenteral antimicrobial therapy (OPAT) is a preferred method to administer long-term intravenous (IV) antibiotic therapy for various infections. OPAT reduces the length of hospital stays and costs while improving patient satisfaction [1,2]. The use of IV antibiotics in an unmonitored setting puts patients at risk for adverse events and hospital readmission. The rate of adverse events ranges from 11.8–63.2% depending on patient factors and the antibiotic used [3,4,5]. Vancomycin and semisynthetic penicillins are typically associated with a higher risk of adverse reactions [3,4,6,7,8,9]. This risk can be mitigated by regular laboratory monitoring, a shorter OPAT duration and frequent outpatient follow ups [1]. It is not always feasible for patients to visit an infusion center or have frequent physician visits. Therefore, monitoring laboratory tests and the use of home nursing visits with regular reporting to outpatient clinics where patients have regular follow ups could be an alternative strategy [10,11].

The IDSA OPAT guidelines on monitoring laboratory results for individual antibiotics are based on known adverse events associated with that antibiotic [1]. The guidelines recommend at least a weekly complete blood count with differential (CBC) and serum creatinine for all parenteral antibiotics. CBCs monitor secondary infections, hypersensitivity drug reactions and drug-related cytopenia. Serum creatinine helps monitor nephrotoxicity and guides appropriate antibiotics dose adjustment. The guidelines recommended monitoring serum electrolytes and liver enzyme levels when certain antibiotics such as beta-lactams are used. A complete metabolic panel (CMP) can monitor renal function, electrolytes, and liver enzyme levels while a basic metabolic panel (BMP) only monitors renal function and electrolyte levels. The guidelines also recommend weekly therapeutic drug monitoring while using aminoglycosides and vancomycin even in patients with stable renal function and serum drug levels [8,9,12]. There are no good data on how often OPAT monitoring laboratory tests should be obtained. However, the guidelines do suggest monitoring patients with fewer comorbidities and short courses of OPAT less frequently. This study was conducted with the objectives of determining the relevance of different laboratory tests for OPAT follow up and identifying predictors of a low risk of OPAT complications that allow for less-frequent laboratory monitoring.

## 2. Results

### 2.1. Baseline Characteristics

After reviewing the medical records of 326 patients, 246 eligible patients were included. A total of 116 patients had OPAT-related complications and 130 patients had no complications. Demographic and clinical data characteristics of the two groups of patients are shown in Table 1. The median age was 55 years (IQR 40–65) and 56% were male. Patients often had multiple comorbidities. Those without OPAT complications were more likely to have no comorbidities. Of the 246, 126 (51.2%) had bone and joint infections, 27 (11%) had CNS infections, 22 (8.9%) had abdominal infections and 21 (8.5%) had endocarditis.

Infections with gram-positive bacteria other than MRSA and polymicrobial infections were the most common infections identified in 91/26 (37%) and 53/246 (21.5) patients, respectively. Beta-lactams with or without vancomycin were the most common antibiotics prescribed. Ceftriaxone and ertapenem were the most frequently used beta-lactams. Two-hundred and nineteen (89%) patients received one or two antibiotics for OPAT and 175/246 (71.1%) were receiving OPAT for 5 weeks or more. The majority of patients (i.e., 109/116, 93.2% of patients with complications and 123/130, 94.6% of patients without complications) received OPAT at home. Seventy-eight (67.3%) patients with complications and 102 (78.5%) without complications followed up in the outpatient clinic within the first 3 weeks of discharge. Anemia was the common baseline laboratory abnormality, seen in 85 (73.3) patients with complications and 86 (66.2%) patients without complications.

### 2.2. OPAT Complications

Table 1 shows the predictors of OPAT complications. Penicillin and semisynthetic penicillin (i.e., ampicillin and nafcillin) had the highest OPAT complications compared to cephalosporins and ertapenem (15/29, 51.7% vs. 41/156, 26.2%). A univariate analysis found that patients were less likely to have an OPAT complication if they had no comorbidities, CCI scores < 5, genitourinary infections, receive a beta-lactam antibiotic without vancomycin, received ceftriaxone or ertapenem, experienced OPAT durations of 2–4 weeks, had a clinic follow up within 1–3 weeks of discharge and no baseline laboratory abnormalities. Conversely, patients with OPAT complications were likely to have CCI scores of five or more, an MRSA infection, received vancomycin, OPAT durations of 5 weeks or more and a clinic follow up after 3 weeks post hospital discharge. Our multivariate regression model found that CCI scores < 5, the use of ceftriaxone and OPAT durations of 2–4 weeks were associated with fewer complications with OPAT. Conversely, CCI scores of five or more and OPAT durations of 5 weeks or more were associated with OPAT complications.

### 2.3. Laboratory Abnormalities and Risk of Readmission to Hospital or Antibiotic Change

There was no patient who had abnormal urinalysis results while receiving OPAT. Table 2 shows a list of laboratory abnormalities, the time of detection of these abnormalities and their association with hospital readmission. The majority of laboratory abnormalities were on CMP or vancomycin troughs. A total of 21 of the 163 (12.9%) laboratory abnormalities were creatinine elevations, which suggested acute kidney injuries, and eleven patients required hospitalization. Nine of those with acute kidney injuries were on vancomycin, two were on vancomycin and two were on on nafcillin. A total of 41 (25.2%) patients developed electrolyte abnormalities, with 15 requiring outpatient supplementation and 25 with no intervention. Five patients with electrolyte abnormalities required readmission to the hospital. We noted eleven incidences of elevated liver enzymes with seven that resolved spontaneously. One patient was readmitted for nafcillin-induced liver injury and three on vancomycin required changing antibiotics. Patients on vancomycin often required dose adjustments for abnormal vancomycin levels. Thirteen patients changed vancomycin to daptomycin due to OPAT complications. A total of 22 patients (8.9%) had CBC abnormalities, and 7 of these 22 patients (31.8%) required an intervention. Eosinophilia was the most common CBC abnormality seen in 14 patients. Only 3/14 (21.4%) patients with eosinophilia were associated with hypersensitivity reactions. There were three incidences of leukopenia, one in a patient who was on nafcillin and two in patients who were on rifampin. One patient was readmitted for acute blood loss anemia. Two of nine (22.2%) patients on daptomycin developed significantly elevated creatine kinase requiring a switch to vancomycin.

We assessed the risk of readmission or antibiotic change based on laboratory abnormalities in our patient population. Fifty-two patients required readmission or antibiotic change (Table 3). In a multivariate analysis that included CCL scores, late clinic follow ups, abnormal CMPs, abnormal CBCs and abnormal vancomycin levels, only abnormal CMPs were associated with hospitalization or changing antibiotics (aOR: 5.7, 95%Cl 2.85–11.24). Abnormal vancomycin troughs and CBC were not significantly associated with a need for hospitalization or change in antibiotics.

## 3. Discussion

OPAT is a health care cost-saving strategy for patients who require long-term parenteral antibiotics [13,14]. Patient satisfaction is high in OPAT users [15] and the successful completion of OPAT requires minimizing adverse events [16]. Therefore, OPAT programs utilize regular laboratory tests to monitor adverse events and manage the adverse events timely. The IDSA OPAT guidelines suggest the use of CBC with differential, BMP and liver profile testing to monitor adverse events [1]. However, the type of laboratory tests used for OPAT monitoring may vary from program to program depending on the resources and type of drugs used [1,17]. Our results showed that urine analysis will not help with monitoring and confirmed the importance of regular laboratory tests, particularly electrolytes, renal function, and liver enzymes, albeit at a different frequency than what was previously suggested to potentially minimize the cost for OPAT programs.

In our study, only 14 (5.7%) patients received OPAT at a skilled nursing facility (SNF). The risk of rehospitalization did not differ between the group who received OPAT at home and those receiving OPAT at SNF. This could be because of the small number of patients who had OPAT at SNF. A prior study showed that patients who received OPAT at SNF had a 46% increased risk of rehospitalization compared to patients who received OPAT at home [18,19].

Our results showed that patients with few comorbidities (CCL < 5), patients who used ceftriaxone alone and patients with an OPAT duration of 4 weeks or less could have laboratory monitoring less frequently than once a week. This study suggests that patients on ceftriaxone are less likely to experience OPAT complications and could require less frequent monitoring. Ertapenem was associated with more complications, but this could be incidental since it was more commonly used with vancomycin. Vancomycin use has a strong association with adverse events, as has been shown in multiple studies, so patients should have at least weekly vancomycin troughs and serum creatinine levels [8,12].

In our study, penicillin and semisynthetic penicillin were the most common causes of abnormal laboratory results. It has been previously reported that semisynthetic penicillins are associated with higher rates of adverse events compared to cephalosporins and ertapenem [4,6,7,20]. In our study, CBC abnormalities were less frequently associated with hospitalization or a change in antibiotics compared to CMP abnormalities. Eosinophilia was the most common CBC abnormality. The 56.9% rate of eosinophilia seen in our OPAT cohort was much higher than the 25% rate of eosinophilia reported previously [21]. While most patients with eosinophilia do not develop an associated clinical abnormality, eosinophilia increases the hazard rate of developing a rash and experiencing renal injury [21]. 

Metronidazole and rifampin were common oral antibiotics used with either vancomycin or beta-lactams for broader coverage and for a better activity in biofilm, respectively. Neither were associated with OPAT complications that warranted hospitalization, but two patients developed leukopenia while on rifampin. Metronidazole is generally safe for use during OPAT, and to our knowledge, there is only one report that showed an unexpected increase in AKI in patients receiving metronidazole, but these patients were also receiving vancomycin which was likely the main cause for AKI [22]. Rifampin, when used during OPAT, is safe, but adverse events arising from interactions with medications for other comorbidities should be monitored [23].

In addition to weekly laboratory follow ups, other different strategies have been employed to decrease rehospitalization rates. It has been reported that regular telephone calls after hospital discharge to ensure adherence is associated with lower readmission rates [24]. Because of the use of central lines and long antibiotic courses, OPAT patients could be at risk of vascular access complications and C-difficile diarrhea. In one study, vascular access complications (i.e., occlusion, line dislodgement, leaking or bleeding, pain and infection) that require intervention have been seen in about 9% of OPAT courses [25]. Long OPAT durations and IVDU have been found to increase the risk of vascular access complications [25]. Despite the perceived risk, it was found that the incidence of Clostridium difficile infection in OPAT patients was low. A single-center study of OPAT patients conducted at Cleveland clinic, USA found that there were 0.26 cases of C-difficile infection per 1000 patient days [26]. Furthermore, OPAT patients who receive antibiotics at home have a significantly lower risk of experiencing C. difficile diarrhea compared to patients who complete similar long-course antibiotics in the hospital [13], indicating that OPAT minimizes hospital-acquired infections.

In our study, early clinic follow ups (i.e., before 3 weeks after hospital discharge) were associated with fewer OPAT complications, whereas late clinic follow ups were associated with more OPAT complications. The time from discharge to clinic follow up is known to have an association with an increased risk of OPAT complications [27]. Patients that would require weekly or more frequent OPAT laboratory monitoring include patients with multiple comorbidities (CCL ≥ 5) and OPAT courses of 5 weeks or more. In one case series, patients with multiple comorbidities tended to have more antibiotic side effects [28]. In fact, some comorbidities such as cardiac and renal failure are considered a relative contraindication to OPAT by some because these comorbidities increase the risk of OPAT failure [29].

## 4. Materials and Methods

Settings: This study was conducted at Saint Louis University Hospital, a 350-bed tertiary academic medical center, over a 7-year interval (1 January 2014 to 31 December 2021). Patients that were discharged home and receiving OPAT received antibiotics from a home infusion company for self-administration. A home health care nurse visited weekly to maintain the peripherally inserted central catheter (PICC) line and obtain samples for weekly laboratory tests. In our center, we routinely order weekly CBCs with differential, CMPs and urine analyses (UA) with microscopy for all OPAT patients. In addition, we order specific serum drug levels for patients on vancomycin or aminoglycoside and creatinine kinase (CK) levels for patients on daptomycin weekly. Infectious disease (ID) fellows review laboratory results regularly and follow patients in the clinic under the supervision of ID attendings.

Data collection: After approval from the Institutional Review Board, we collected clinical information from our institution’s electronic medical records. The data collected included age, gender, comorbidities, indications for OPAT, microbiology, prescribed antibiotics, OPAT duration, discharge location, time from discharge to outpatient clinic follow up, baseline laboratory abnormalities and the type and number of abnormal laboratory results while receiving OPAT. If patients were readmitted while receiving OPAT, we documented the date and reason for hospitalization. If the physician changed antibiotics, we documented the date of the change and reasons for the change.

Inclusion and exclusion criteria: The patients included in this study were 18 years old or older, required two weeks or more of OPAT and was evaluated by an infectious diseases consult team while in the hospital. We excluded those that entered hospice care, had missing data and were readmitted for more than one week for causes not directly related to antibiotic treatment or abnormal OPAT follow-up results.

Definitions and Variables: We separated patients into a cohort of OPAT patients with complications and patients with no complications. We defined an OPAT complication as a clinically significant laboratory abnormality, a need to change antibiotic class and hospital readmission for any reason. We defined immunosuppression as chronic glucocorticoid use, acquired immunodeficiency syndrome with CD4 < 200/mL, use of disease-modifying antirheumatic drugs (DMARD), solid organ transplant or bone marrow transplant. We used the Charlson comorbidity index (CCI) to estimate a 10-year mortality risk. The OPAT duration was the number of weeks from hospital discharge until the physician discontinued antibiotics. Laboratory values at initial discharge from the hospital were considered baseline. Clinically significant laboratory values included creatinine, sodium, potassium, aspartate aminotransferase, alanine transaminase, alkaline phosphatase, total white blood count, platelet count, absolute eosinophil count, vancomycin trough, CK and urinalysis.

Statistical Analysis: Continuous variables are reported in medians and interquartile ranges (IQRs). Nominal and categorical variables are reported as proportions and frequencies. Univariate analysis with Chi square test of independence was used to identify the risk factors of OPAT complications. Variables with a *p* value < 0.05 from the univariate analysis were included for the multivariable logistic regression model. We considered a *p* value  <  0.05 statistically significant for our regression model. IBM SPSS 26 was used for the analysis.

## 5. Conclusions

Complications are not uncommon in OPAT patients. Recognizing predictors of OPAT complications will help determine strategies to optimize care, increase patient satisfaction and potentially further decrease cost. The less frequent monitoring of laboratory values could be useful in OPAT patients at a lower risk of complications. Less-frequent laboratory testing could be important to reduce costs (e.g., nursing effort to draw blood, laboratory cost and physician time to follow results) and potentially increase patient satisfaction (e.g., less blood draw).

## Figures and Tables

**Table 1 antibiotics-12-00330-t001:** Demographics of patients with or without any OPAT complications.

	OPAT Complications (n = 116)	No OPAT Complications (n = 130)	*p* Value	Multivariate Analysis aOR	Multivariate Analysis 95% Cl
Age (median, IQR)	54.5 (40–65)	55 (42–65)			
Gender (female)	53 (45.7%)	56 (43.1%)	0.68		
Comorbidities					
Cardiovascular disease	24	21	0.36		
Liver cirrhosis	7	6	0.62		
CKD/ESRD	12	13	0.93		
DM	35	35	0.57		
Malignancy	12	13	0.93		
Immunosuppression	13	22	0.2		
Obesity	19	21	0.96		
None	27	47	0.03	0.9	0.43–1.9
Carlson Comorbidity Index					
0–4	88 (75.9%)	115 (88.5%)	0.009	0.39	0.2–0.77
≥5	28 (24.1%)	15 (11.5%)	0.009	2.21	1.11–4.43
Indications for OPAT					
Abdominal Infection	9 (7.8%)	13 (10.0%)	0.54		
CNS infection	16 (13.8%)	11 (8.5%)	0.18		
Endocarditis	10 (8.6%)	11 (8.5%)	0.61		
Vascular Infection	8 (6.9%)	7 (5.4%)	0.62		
ENT infection	1 (0.9%)	6 (4.6%)	0.08		
Genitourinary Infection	1 (0.9%)	10 (7.7%)	0.01	0.1	0.01–1.01
Bone and Joint infections	67 (57.8%)	59 (45.4%)	0.052	1.1	0.6–2.1
Skin–Soft Tissue Infection	2 (1.7%)	5 (0.32%)	0.32		
Pulmonary Infection	2 (1.7%)	9 (6.9%)	0.06		
Pathogen treated					
Gram-positive Bacteria	40 (34.5%)	51 (39.2%)	0.44		
Gram-negative Bacteria	11 (9.5%)	11 (8.5%)	0.78		
MRSA	22 (19.0%)	9 (6.9%)	0.004	2.4	0.82–7.0
Pseudomonas	1 (0.9%)	7 (5.4%)	0.06		
MDR-gram negatives	3 (2.6%)	11 (8.5%)	0.06		
Candida	0	1 (0.8%)	0.34		
Polymicrobial	25 (21.6%)	28 (21.5%)	0.98		
None	14 (12.1%)	12 (9.2%)	0.47		
Antibiotic class (es)					
Beta-lactams w/o vancomycin	57 (49.1%)	98 (75.4%)	<0.001	1.3	0.28–6.3
Vancomycin + others	56 (48.3%)	26 (20.0%)	<0.002	2.8	0.57–13.9
Daptomycin	3 (2.6%)	6 (4.6%)	0.4		
Beta-lactams used					
Aztreonam	1	0	0.3		
Cefazolin	4	3	0.6		
Ceftriaxone	15	60	<0.001	0.47	0.24–0.91
Cefepime	17	17	0.72		
Ertapenem	5	35	<0.001	0.6	0.24–1.4
Penicillin G	2	6	0.2		
Ampicillin	3	0	0.07		
Nafcillin	10	8	0.46		
Piperacillin/Tazobactam	5	3	0.38		
Aminoglycoside	0	3	0.1		
Fluoroquinolone	3	0	0.07		
metronidazole	31	30	0.51		
Rifampin	6	4	0.41		
Number of antibiotics					
1	60 (51.7%)	79 (60.8%)	0.15		
2	35 (30.2%)	45 (34.6%)	0.46		
3	21 (18.1%)	6 (4.6%)	0.0007	2.0	0.7–5.9
Duration of OPAT (weeks)					
2–4	20 (17.2%)	51 (39.2%)	0.0001	0.37	0.23–0.77
≥5	96 (82.8%)	79 (60.8%)	0.0001	2.7	1.3–5.6
Discharge Location	\				
Home	109 (93.2%)	123 (94.6%)	0.83		
SNF	7 (6.8%)	7 (5.4%)	0.83		
Weeks from Discharge to Clinic					
None	18 (15.5%)	20 (15.4%)	0.98		
1 to 3	78 (67.3%)	102 (78.5%)	0.047	0.54	0.23–1.2
4 to 6	20 (17.2%)	8 (6.1%)	0.006	1.4	0.41–4.7
Baseline Abnormal lab					
Anemia	85	86	0.22		
Eosinophilia	2	2	0.91		
Leukopenia	4	3	0.59		
Thrombocytopenia	6	5	0.61		
Resolving AKI	4	7	0.46		
Transaminitis	6	2	0.11		
None	21	39	0.03	0.62	0.3–1.3

**Table 2 antibiotics-12-00330-t002:** Total number of OPAT complications.

Abnormal Labs		Hospitalization		Change in Antibiotic Class	
CMP	73	AKI	11	vancomycin	13
CBC	22	Electrolyte abnormality	5	nafcillin	5
Vancomycin Trough	61	Drug Liver Injury	1	cefepime	4
Creatine Kinase	2	Anemia	1	rifampin	2
		Drug rash on Labs	2	daptomycin	2
		Drug rash not on labs	3	ampicillin	1
		PICC line thrombosis	6		
		New Infection	8		
		Not OPAT	5		
Total	168	Total	42	Total	27

**Table 3 antibiotics-12-00330-t003:** Laboratory abnormalities and risk of hospital readmission while receiving OPAT.

	Hospitalized/Changed Antibiotic (N =52)	No Change or Hospitalization (n = 194)	Univariate Analysis *p* Value	aOR	95% Cl
Abnormal lab results (total)	66	97			
CMP	37	41	<0.0001	3.3	1.5–7.4
Abnormal serum creatinine	15	7	<0.0001		
Electrolyte abnormality	17	28	0.002		
Elevated transaminases	5	6	0.04		
CBC	7	15	0.2		
Eosinophilia	3	11	0.98		
Leukopenia	2	3	0.3		
Anemia	2	1	0.052		
High/low vanc level	20	41	0.01	0.867	0.34–2.2
Elevated CK	2	0	0.008		
Abnormal lab by week					
1	10	10	0.001	5.1	1.4–19.3
2	30	24	<0.0001	4.02	1.5–10.8
3	8	18	0.2		
4	11	20	0.01	2.2	0.67–7.04
5	3	17	0.48		
6	5	9	0.17		

## Data Availability

Data will be available upon request.

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
