# Peer review of "Usefulness of Routine Laboratory Tests for Follow up of Patients Receiving Outpatient Parenteral Antimicrobial Therapy Run by Infectious Diseases Fellows"

_antibiotics, 2023, doi:10.3390/antibiotics12020330_

Round 1

Reviewer 1 Report

Make abstract structured

These details about your study is too much to include in the introduction section. Move details to method and make your objective brief

Avoid abbreviations at beginning of sentences 

Make subsections in the method section

Author Response

  1. Make abstract structured

Abstract structured into background, methods, results and conclusion

  1. These details about your study is too much to include in the introduction section. Move details to method and make your objective brief.

Part of the methods section is moved to the Method section and objectives made brief.

  1. Avoid abbreviations at beginning of sentences 

We have avoided abbreviations at the beginning of sentences. Abbreviations were used only after full definition the first time they were used.

  1. Make subsections in the method section

The method section has five subsections: setting, data collection, definition and variables, and statistical analysis.

Reviewer 2 Report

Please review and comment on the following references:

On Clostridioides difficile:

Wong KK.

Infect Control Hosp Epidemiol 2015; 36:110.

Staples JA.

Clin Infect Dis 2022; 75:1921.

On adverse events and costs:

Staples JA.

Clin Infect Dis 2022; 75:1921.

On follow-up and automated voice calls:

Huggings CE.

Hosp Pharm 2022; 57:107.

Other antibiotics:

Hughes C.

JAC Antimicrob Resist 2019; 1:dzl085

Complications:

Shrestha NK.

J Antimicrob Chemother 2016; 71:506.

Author Response

Please review and comment on the following references:

 On Clostridioides difficile:

  • Wong KK. Infect Control Hosp Epidemiol 2015; 36:110.
  • Staples JA. Clin Infect Dis 2022; 75:1921.

On adverse events and costs:

  • Staples JA. Clin Infect Dis 2022; 75:1921.

 On follow-up and automated voice calls:

  • Huggings CE. Hosp Pharm 2022; 57:107.

 Other antibiotics:

  • Hughes C. JAC Antimicrob Resist 2019; 1:dzl085

 Complications:

  • Shrestha NK. J Antimicrob Chemother 2016; 71:506.

We have reviewed all references and we have included the new references in the discussion section.

Reviewer 3 Report

Jared F et al. in their paper presented a retrospective observational cohort study over the course of 7-year interval in order to identify risk factors associated with abnormal laboratory test and to determine if abnormal laboratory tests led to antibiotic change or hospital readmission.

Overall the manuscript is well written, presented and does tackle the important medical issue of outpatient parenteral antimicrobial therapy. In a multivariate analysis, the paper analyzed the association of comorbidity factor and recommendations are made to optimize care, increase patient satisfaction and potentially decrease cost.

Author Response

  1. English language and style are fine/minor spell check required

Spell checked.

  1. Jared F et al. in their paper presented a retrospective observational cohort study over the course of 7-year interval in order to identify risk factors associated with abnormal laboratory test and to determine if abnormal laboratory tests led to antibiotic change or hospital readmission. Overall the manuscript is well written, presented and does tackle the important medical issue of outpatient parenteral antimicrobial therapy. In a multivariate analysis, the paper analyzed the association of comorbidity factor and recommendations are made to optimize care, increase patient satisfaction and potentially decrease cost.

We appreciate the reviewer for comment.

Reviewer 4 Report

The paper describes the experience of the Saint Louis University Hospital in the monitoring of outpatient parenteral antimicrobial therapy (OPAT).

The results are of interest to the reader and may be useful in optimizing the management of these patients.

Comments: page 10, inclusion and exclusion criteria: why were patients readmitted for more than one week while on OPAT excluded?

Author Response

  1. English language and style are fine/minor spell check require

Spell checked.

  1. The paper describes the experience of the Saint Louis University Hospital in the monitoring of outpatient parenteral antimicrobial therapy (OPAT). The results are of interest to the reader and may be useful in optimizing the management of these patients.

We appreciate the reviewer for comment.

  1. Comments: page 10, inclusion and exclusion criteria: why were patients readmitted for more than one week while on OPAT excluded?

Readmission for more than one week was considered as an exclusion criterion because it may potentially lead to missed weekly laboratory values. However, this was only for those who were admitted for reasons not directly related to antibiotics or follow laboratory abnormalities. Therefore, the statement is now rephrased to give a better explanation: ‘We excluded patients readmitted for more than one week for causes not directly related to antibiotic treatment or abnormal follow up OPAT laboratories.